# Identification of novel, clinically correlated autoantigens in the monogenic autoimmune syndrome APS1 by proteome-wide PhIP-Seq

Sara E Vazquez[1,2,3,4], Elise MN Ferré[5], David W Scheel[3], Sara Sunshine[4,6], Brenda Miao[3], Caleigh Mandel-Brehm[4], Zoe Quandt[7], Alice Y Chan[8], Mickie Cheng[3], Michael German[3,7,9], Michail Lionakis[5], Joseph L DeRisi[4,10†]*, Mark S Anderson[3,7†]*

[1]Medical Scientist Training Program, University of California, San Francisco, San Francisco, United States; [2]Tetrad Graduate Program, University of California, San Francisco, San Francisco, United States; [3]Diabetes Center, University of California, San Francisco, San Francisco, United States; [4]Department of Biochemistry and Biophysics, University of California, San Francisco, San Francisco, United States; [5]Fungal Pathogenesis Section, Laboratory of Clinical Immunology and Microbiology, National Institute of Allergy & Infectious Diseases, National Institutes of Health, Bethesda, United States; [6]Biomedical Sciences Graduate Program, University of California, San Francisco, San Francisco, United States; [7]Department of Medicine, University of California, San Francisco, San Francisco, United States; [8]Department of Pediatrics, University of California, San Francisco, San Francisco, United States; [9]Eli and Edythe Broad Center of Regeneration Medicine and Stem Cell Research, University of California, San Francisco, San Francisco, United States; [10]Chan Zuckerberg Biohub, San Francisco, United States

*For correspondence:
joe@derisilab.ucsf.edu (JLDR);
Mark.Anderson@ucsf.edu (MSA)

†These authors contributed equally to this work

**Abstract** The identification of autoantigens remains a critical challenge for understanding and treating autoimmune diseases. Autoimmune polyendocrine syndrome type 1 (APS1), a rare monogenic form of autoimmunity, presents as widespread autoimmunity with T and B cell responses to multiple organs. Importantly, autoantibody discovery in APS1 can illuminate fundamental disease pathogenesis, and many of the antigens found in APS1 extend to more common autoimmune diseases. Here, we performed proteome-wide programmable phage-display (PhIP-Seq) on sera from a cohort of people with APS1 and discovered multiple common antibody targets. These novel APS1 autoantigens exhibit tissue-restricted expression, including expression in enteroendocrine cells, pineal gland, and dental enamel. Using detailed clinical phenotyping, we find novel associations between autoantibodies and organ-restricted autoimmunity, including a link between anti-KHDC3L autoantibodies and premature ovarian insufficiency, and between anti-RFX6 autoantibodies and diarrheal-type intestinal dysfunction. Our study highlights the utility of PhIP-Seq for extensively interrogating antigenic repertoires in human autoimmunity and the importance of antigen discovery for improved understanding of disease mechanisms.

## Introduction

Autoimmune Polyglandular syndrome type 1 (APS1) or Autoimmune Polyglandular-Candidiasis-Ectodermal Dystrophy (APECED; OMIM #240300) is an autoimmune syndrome caused by monogenic

**eLife digest** The immune system uses antibodies to fight microbes that cause disease. White blood cells pump antibodies into the bloodstream, and these antibodies latch onto bacteria and viruses, targeting them for destruction. But sometimes, the immune system gets it wrong. In autoimmune diseases, white blood cells mistakenly make antibodies that target the body's own tissues. Detecting these 'autoantibodies' in the blood can help doctors to diagnose autoimmune diseases. But the identities and targets of many autoantibodies remain unknown.

In one rare disease, called autoimmune polyendocrine syndrome type 1 (APS-1), a faulty gene makes the immune system much more likely to make autoantibodies. People with this disease can develop an autoimmune response against many different healthy organs. Although APS-1 is rare, some of the autoantibodies made by individuals with the disease are the same as the ones in more common autoimmune diseases, like type 1 diabetes. Therefore, investigating the other autoantibodies produced by individuals with APS-1 could reveal the autoantibodies driving other autoimmune diseases.

Autoantibodies bind to specific regions of healthy proteins, and one way to identify them is to use hundreds of thousands of tiny viruses in a technique called proteome-wide programmable phage-display, or PhIP-Seq. Each phage carries one type of protein segment. When mixed with blood serum from a patient, the autoantibodies stick to the phages that carry the target proteins for that autoantibody. These complexes can be isolated using biochemical techniques. Sequencing the genes of these phages then reveals the identity of the autoantibodies' targets.

Using this technique, Vazquez et al successfully pulled 23 known autoantibodies from the serum of patients with APS-1. Then, experiments to search for new targets began. This revealed many new autoantibodies, targeting proteins found only in specific tissues. They included one that targets a protein found on cells in the gut, and another that targets a protein found on egg cells in the ovaries. Matching the PhIP-Seq data to patient symptoms confirmed that these new antibodies correlate with the features of specific autoimmune diseases. For example, patients with antibodies that targeted the gut protein were more likely to have gut symptoms, while patients with antibodies that targeted the egg cell protein were more likely to have problems with their ovaries.

Further investigations using PhIP-Seq could reveal the identities of even more autoantibodies. This might pave the way for new antibody tests to diagnose autoimmune diseases and identify tissues at risk of damage. This could be useful not only for people with APS-1, but also for more common autoimmune diseases that target the same organs.

mutations in the *AIRE* gene that result in defects in AIRE-dependent T cell education in the thymus (*Aaltonen et al., 1997*; *Anderson et al., 2002*; *Conteduca et al., 2018*; *Malchow et al., 2016*; *Nagamine et al., 1997*). As a result, people with APS1 develop autoimmunity to multiple organs, including endocrine organs, skin, gut, and lung (*Ahonen et al., 1990*; *Ferre et al., 2016*; *Söderbergh et al., 2004*). Although the majority of APS1 autoimmune manifestations are thought to be primarily driven by autoreactive T cells, people with APS1 also possess autoreactive B cells and corresponding high-affinity autoantibody responses (*Devoss et al., 2008*; *Gavanescu et al., 2008*; *Meyer et al., 2016*; *Sng et al., 2019*). These autoantibodies likely derive from germinal center reactions driven by self-reactive T cells, resulting in mirroring of autoantigen identities between the T and B cell compartments (*Lanzavecchia, 1985*; *Meyer et al., 2016*).

Identification of the specificity of autoantibodies in autoimmune diseases is important for understanding underlying disease pathogenesis and for identifying those at risk for disease (*Rosen and Casciola-Rosen, 2014*). However, despite the long-known association of autoantibodies with specific diseases in both monogenic and sporadic autoimmunity, many autoantibody specificities remain undiscovered. Challenges in antigen identification include the weak affinity of some autoantibodies for their target antigen, as well as rare or low expression of the target antigen. One approach to overcome some of these challenges is to interrogate autoimmune patient samples with particularly high affinity autoantibodies. Indeed, such an approach identified GAD65 as a major autoantigen in type 1 diabetes by using sera from people with Stiff Person Syndrome (OMIM #184850), who harbor high affinity autoantibodies (*Baekkeskov et al., 1990*). We reasoned that PhIP-Seq interrogation of

APS1, a defined monogenic autoimmune syndrome with a broad spectrum of high affinity autoantibodies, would likely yield clinically meaningful targets – consistent with previously described APS1 autoantibody specificities that exhibit strong, clinically useful associations with their respective organ-specific diseases (*Alimohammadi et al., 2008*; *Alimohammadi et al., 2009*; *Ferré et al., 2019*; *Landegren et al., 2015*; *Popler et al., 2012*; *Puel et al., 2010*; *Shum et al., 2013*; *Söderbergh et al., 2004*; *Winqvist et al., 1993*).

The identification of key B cell autoantigens in APS1 has occurred most commonly through candidate-based approaches and by whole-protein microarrays. For example, lung antigen BPIFB1 autoantibodies, which are used to assess people with APS1 for risk of interstitial lung disease, were discovered first in *Aire*-deficient mice using a combination of targeted immunoblotting, tissue microscopy, and mass spectrometry (*Shum et al., 2009*; *Shum et al., 2013*). Recently, there have been rapid advances in large platform approaches for antibody screening; these platforms can overcome problems of antigen abundance by simultaneously screening the majority of proteins from the human genome in an unbiased fashion (*Jeong et al., 2012*; *Larman et al., 2011*; *Sharon and Snyder, 2014*; *Zhu et al., 2001*). In particular, a higher-throughput antibody target profiling approach utilizing a fixed protein microarray technology (ProtoArray) has enabled detection of a wider range of proteins targeted by autoantibodies directly from human serum (*Fishman et al., 2017*; *Landegren et al., 2016*; *Meyer et al., 2016*). Despite initial success of this technology in uncovering shared antigens across APS1 cohorts, it is likely that many shared antigens remain to be discovered, given that these arrays do not encompass the full coding potential of the proteome.

Here, we took an alternate approach to APS1 antigen discovery by employing Phage Immunoprecipitation-Sequencing (PhIP-Seq) based on an established proteome-wide tiled library (*Larman et al., 2011*; *O'Donovan et al., 2018*). This approach possesses many potential advantages over previous candidate-based and whole-protein fixed array approaches, including (1) expanded, proteome-wide coverage (including alternative splice forms) with 49 amino acid (AA) peptide length and 24AA resolution tiling, (2) reduced volume requirement for human serum, and (3) high-throughput, sequencing based output (*Larman et al., 2011*; *O'Donovan et al., 2018*). Of note, the PhIP-Seq investigation of autoimmune diseases of the central nervous system, including paraneoplastic disease, has yielded novel and specific biomarkers of disease (*Larman et al., 2011*; *Mandel-Brehm et al., 2019*; *O'Donovan et al., 2018*).

Using a PhIP-Seq autoantibody survey, we identify a collection of novel APS1 autoantigens as well as numerous known, literature-reported APS1 autoantigens. We orthogonally validate seven novel autoantigens including RFX6, KHDC3L, and ACP4, all of which exhibit tissue-restricted expression (*Patel et al., 2017*; *Rezaei et al., 2016*; *Seymen et al., 2016*; *Smith et al., 2017*; *Smith et al., 2010*; *Zhu et al., 2015*). Importantly, these novel autoantigens may carry important implications for poorly understood clinical manifestations such as intestinal dysfunction, ovarian insufficiency, and tooth enamel hypoplasia, where underlying cell-type specific antigens have remained elusive. Together, our results demonstrate the applicability of PhIP-Seq to antigen discovery, substantially expand the spectrum of known antibody targets and clinical associations in APS1, and point towards novel specificities that can be targeted in autoimmunity.

## Results

### Investigation of APS1 serum autoantibodies by PhIP-Seq

Individuals with APS1 develop autoantibodies to many known protein targets, some of which exhibit tissue-restricted expression and have been shown to correlate with specific autoimmune disease manifestations. However, the target proteins for many of the APS1 tissue-specific manifestations remain enigmatic. To this end, we employed a high-throughput, proteome-wide programmable phage display approach (PhIP-Seq) to query the antibody target identities within serum of people with APS1 (*Larman et al., 2011*; *O'Donovan et al., 2018*). The PhIP-Seq technique leverages large scale oligo production and efficient phage packaging and expression to present a tiled-peptide representation of the proteome displayed on T7 phage. Here, we utilize a phage library that we previously designed and deployed for investigating paraneoplastic autoimmune encephalitis (*Mandel-Brehm et al., 2019*; *O'Donovan et al., 2018*). The library itself contains approximately 700,000 unique phage, each displaying a 49 amino acid proteome segment. As previously described, phage

were immunoprecipitated using human antibodies bound to protein A/G beads. In order to increase sensitivity and specificity for target proteins, eluted phage were used for a further round of amplification and immunoprecipitation. DNA was then extracted from the final phage elution, amplified and barcoded, and subjected to Next-Generation Sequencing (*Figure 1A*). Finally, sequencing counts were normalized across samples to correct for variability in sequencing depth, and the fold-change of each gene was calculated (comprised of multiple unique tiling phage) as compared to mock IPs in the absence of human serum (further details of the protocol can be found in the methods section).

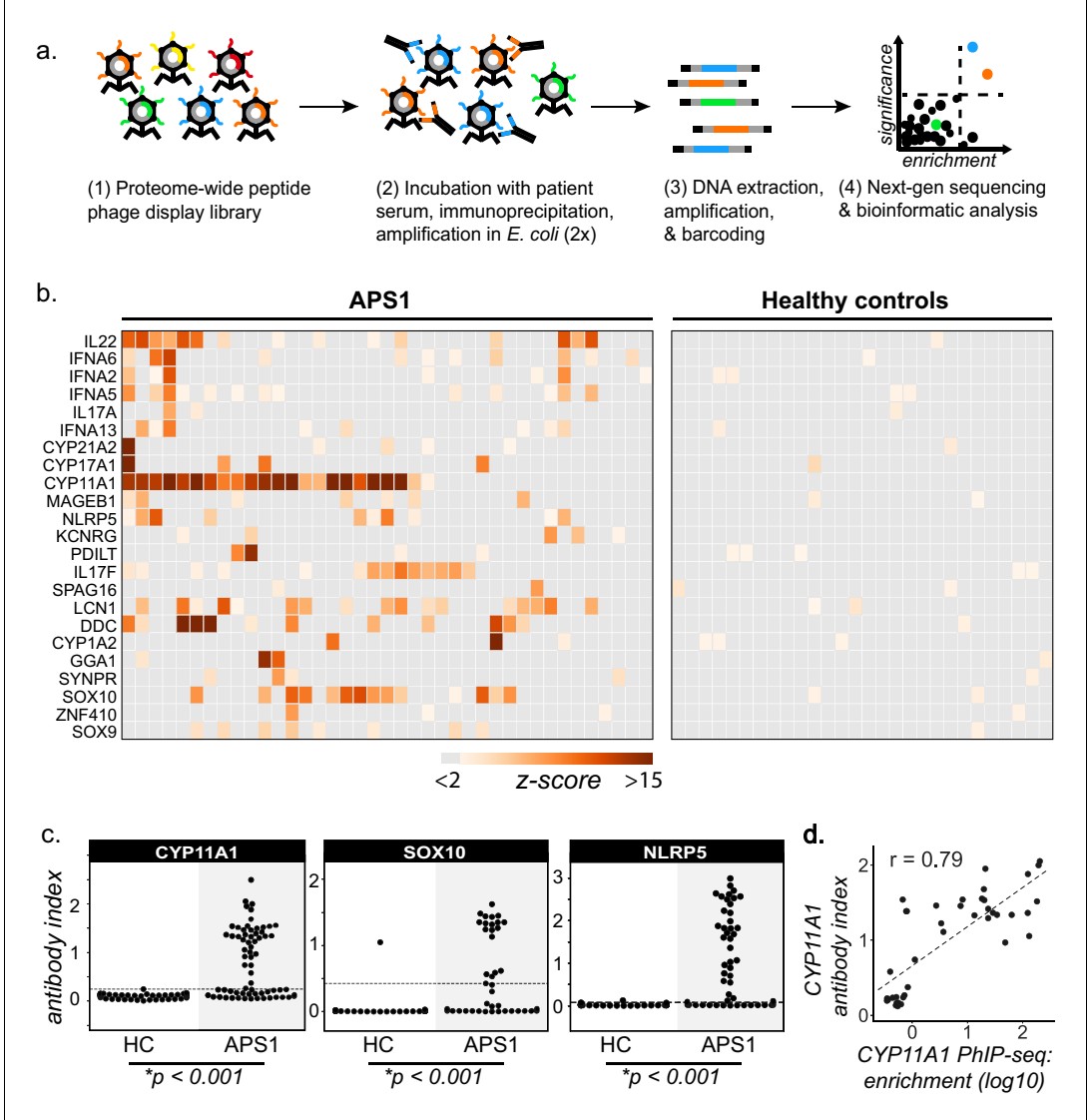

**Figure 1.** PhIP-Seq identifies literature-reported autoantigens in APS1. (A) Overview of PhIP-Seq experimental workflow. (B) PhIP-Seq identifies known autoantibody targets in APS1. Hierarchically clustered (Pearson) z-scored heatmap of literature reported autoantigens with 10-fold or greater signal over mock-IP in at least 2/39 APS1 sera and in 0/28 non-APS1 control sera. (C) Radioligand binding assay (RLBA) orthogonal validation of literature-reported antigens CYP11A1, SOX10, and NLRP5 within the expanded cohort of APS1 (n = 67) and non-APS1 controls (n = 61); p-value was calculated across all samples using a Mann-Whitney U test. Dashed line indicates mean of healthy control signal + 3 standard deviations. (D) CYP11A1 RLBA antibody index and CYP11A1 PhIP-Seq enrichment are well correlated (Pearson, r = 0.79).

The online version of this article includes the following figure supplement(s) for figure 1:

**Figure supplement 1.** Hierarchically clustered (Pearson) z-scored heatmap of literature reported autoantigens that did not meet the cutoff of 10-fold or greater signal over mock-IP in at least 2/39 APS1 sera and in 0/28 non-APS1 control sera.

**Figure supplement 2.** Additional PhIP-Seq data for known autoantigens SOX10 and NLRP5.

From a cohort of 67 APS1 serum samples, a total of 39 samples were subjected to PhIP-Seq investigation, while the remaining 28 samples were obtained at a later time point and reserved for downstream validation experiments (for clinical data, refer to *Supplementary file 1*). In addition, 28 non-APS1 anonymous blood donor serum samples were subjected to PhIP-Seq, and an additional group of 61 non-APS1 plasma samples were used for downstream validation experiments (*Supplementary file 2*).

## Detection of literature-reported APS1 autoantigens

PhIP-Seq results were first cross-referenced with previously reported APS1 autoantibody targets (*Alimohammadi et al., 2008*; *Alimohammadi et al., 2009*; *Clemente et al., 1997*; *Fishman et al., 2017*; *Hedstrand et al., 2001*; *Husebye et al., 1997*; *Kluger et al., 2015*; *Kuroda et al., 2005*; *Landegren et al., 2015*; *Landegren et al., 2016*; *Leonard et al., 2017*; *Meager et al., 2006*; *Meyer et al., 2016*; *Oftedal et al., 2015*; *Pöntynen et al., 2006*; *Sansom et al., 2014*; *Shum et al., 2009*; *Shum et al., 2013*; *Söderbergh et al., 2004*). To avoid false positives, a conservative set of criteria were used as follows. We required a minimum of 2/39 APS1 samples and 0/28 non-APS1 control samples to exhibit normalized gene counts in the immunoprecipitation (IP) with greater than 10-fold enrichment as compared to the control set of 17 mock-IP (beads, no serum) samples. This simple, yet stringent criteria enabled detection of a total of 23 known autoantibody specificities (*Figure 1B*). Importantly, many of the well-validated APS1 antigens, including specific members of the cytochrome P450 family (CYP1A2, CYP21A1, CYP11A1, CYP17A1), lung disease-associated antigen KCRNG, as well as IL17A, IL17F, and IL22, among others were well represented (*Figure 1B*). In contrast, the diabetes-associated antigens GAD65 and INS did not meet these stringent detection criteria and only weak signal was detected to many of the known interferon autoantibody targets known to be present in many people with APS1, perhaps due to the conformational nature of these autoantigens (*Figure 1B* and *Figure 1—figure supplement 1*; *Björk et al., 1994*; *Meager et al., 2006*; *Meyer et al., 2016*; *Wolff et al., 2013*; *Ziegler et al., 1996*).

Three known autoantigens that were prevalent within our cohort were selected to determine how PhIP-Seq performed against an orthogonal whole protein-based antibody detection assay. A radioligand binding assay (RLBA) was performed by immunoprecipitating in vitro transcribed and translated S35-labeled proteins CYP11A1, SOX10, and NLRP5 with APS1 serum (*Alimohammadi et al., 2008*; *Berson et al., 1956*; *Hedstrand et al., 2001*; *Winqvist et al., 1993*). Importantly, and in contrast to PhIP-Seq, this assay tests for antibody binding to full-length protein (*Figure 1C*). By RLBA, these three antigens were present in and specific to both the initial discovery APS1 cohort (n = 39) as well as the expanded validation cohort (n = 28), but not the non-APS1 control cohort (n = 61). Together, these results demonstrate that PhIP-Seq detects known APS1 autoantigens and that PhIP-Seq results validate well in orthogonal whole protein-based assays.

To determine whether the PhIP-Seq APS1 dataset could yield higher resolution information on antigenic peptide sequences with respect to previously reported targets, the normalized enrichments of all peptides belonging to known disease-associated antigens CYP11A1 and SOX10 were mapped across the full length of their respective proteins (*Figure 1—figure supplement 2*). The antigenic regions within these proteins were observed to be similar across all samples positive for anti-CYP11A1 and anti-SOX10 antibodies, respectively (*Figure 1—figure supplement 2*) suggesting peptide-level commonalities and convergence among the autoreactive antibody repertoires across individuals. These data suggest that people with APS1 often target similar, but not identical protein regions.

## Identification of novel APS1 autoantigens

Having confirmed that PhIP-Seq analysis of APS1 sera detected known antigens, the same data were then investigated for the presence of novel, previously uncharacterized APS1 autoantigens. We applied the same positive hit criteria as described for known antigens, and additionally increased the required number of positive APS1 samples to 3/39 to impose a stricter limit on the number of novel candidate autoantigens. This yielded a list of 81 genes, which included 12 known antigens and 69 putative novel antigens (*Figure 2*).

The most commonly held hypotheses regarding the nature and identity of proteins targeted by the aberrant immune response in APS1 are that targeted proteins (1) tend to exhibit AIRE-

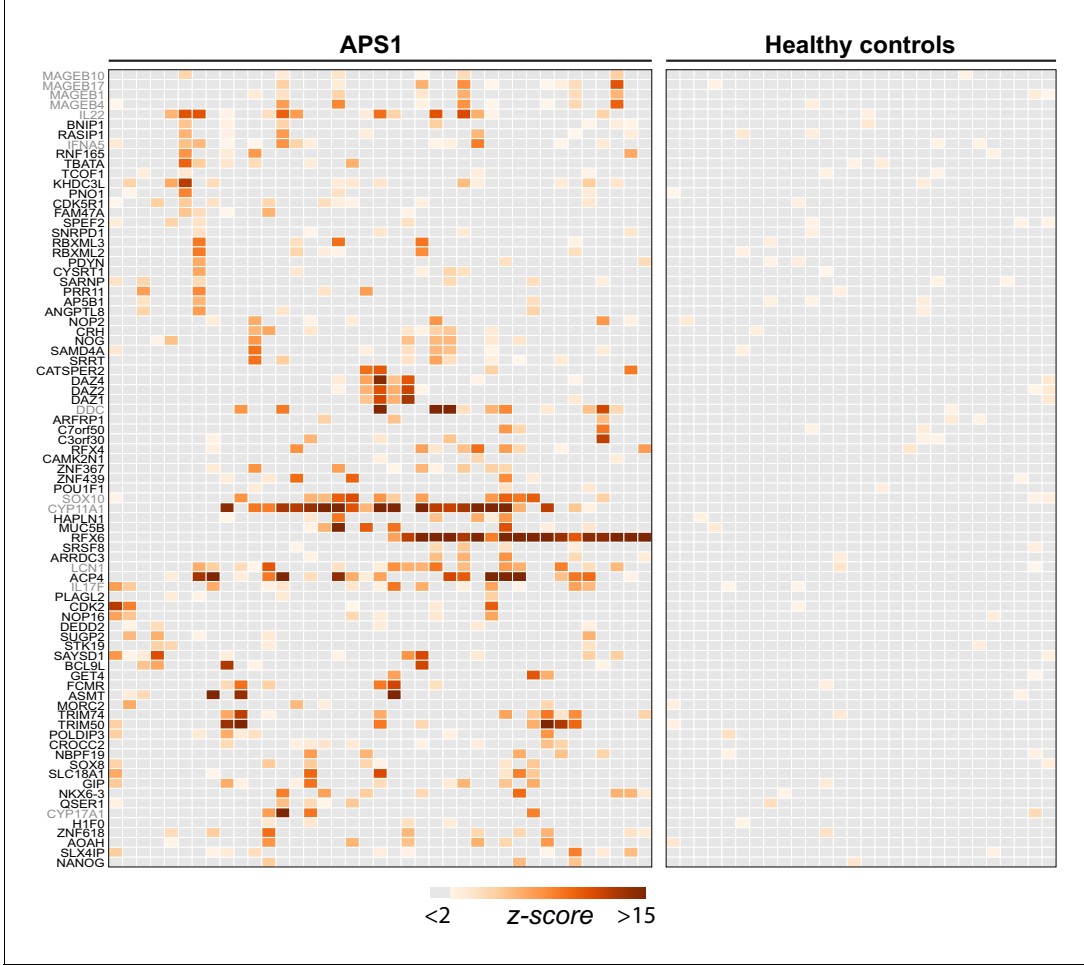

**Figure 2.** PhIP-Seq identifies novel (and known) antigens across multiple APS1 sera. (**A**) Hierarchically clustered (Pearson) z-scored heatmap of all genes with 10-fold or greater signal over mock-IP in at least 3/39 APS1 sera and in 0/28 non-APS1 sera. Black labeled antigens (n = 69) are potentially novel and grey labeled antigens (n = 12) are previously literature-reported antigens.

The online version of this article includes the following figure supplement(s) for figure 2:

**Figure supplement 1.** The mean of tissue-specificity ratio of 81 PhIP-Seq antigens (*Figure 2*) is increased as compared to the tissue-specificity ratio of n = 81 randomly sampled genes (n-sampling = 10'000).

dependent thymic expression and (2) have restricted expression to one or few peripheral organs and tend not to be widely or ubiquitously expressed. We investigated whether our novel antigens were also preferentially tissue-restricted. In order to systematically address this question, tissue-specific RNA expression was assessed using a consensus expression dataset across 74 cell types and tissues (*Uhlén et al., 2015*). For each gene, the ratio of expression in the highest tissue as compared to the sum of expression across all tissues was calculated, resulting in higher ratios for those mRNAs with greater degrees of tissue-restriction. Using this approach, the mean tissue-specificity ratio of the 81 PhIP-Seq positive antigens was increased by approximately 1.5-fold (p=0.0017) as compared to the means from iterative sampling of 81 genes (*Figure 2—figure supplement 1*).

## Identification of novel antigens common to many individuals

Identified autoantigens were ranked by frequency within the cohort. Five antigens were positive in ten or more APS1 samples, including two novel antigens. In addition, the majority of antigens found in 4 or more APS1 sera were novel (*Figure 3A*). Five of the most frequent novel antigens were selected for subsequent validation and follow-up. These included RFX6, a transcription factor implicated in pancreatic and intestinal pathology (*Patel et al., 2017*; *Smith et al., 2010*); ACP4, an enzyme implicated in dental enamel hypoplasia (*Choi et al., 2016*; *Seymen et al., 2016*;

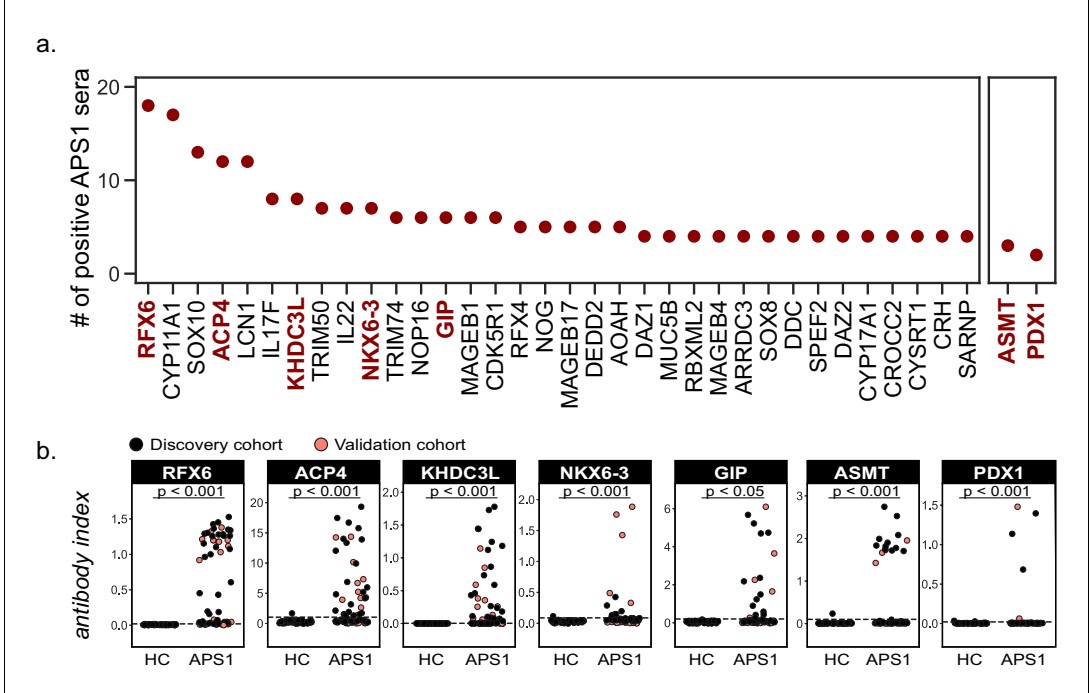

**Figure 3.** Novel PhIP-Seq autoantigens are shared across multiple APS1 samples and validate in whole protein binding assays. (A) Graph of the PhIP-seq autoantigens from **Figure 2** that were shared across the highest number of individual APS1 sera (left panel). ASMT and PDX1 were positive hits in 3 and 2 sera, respectively, but are known to be highly tissue specific (right panel). Genes in red were chosen for validation in whole protein binding assay. (B) Validation of novel PhIP-Seq antigens by radiolabeled binding assay, with discovery cohort (black, $n_{APS1}$ = 39), validation cohort (light red, $n_{APS1}$ = 28) and non-APS1 control cohort ($n_{HC}$ = 61). P-value was calculated across all samples using a Mann-Whitney U test. Dashed line indicates mean of healthy control signal + 3 standard deviations.

The online version of this article includes the following figure supplement(s) for figure 3:

**Figure supplement 1.** Comparison of PhIP-Seq data to orthonongal whole-protein binding assays.

**Smith et al., 2017**); KHDC3L, a protein with oocyte-restricted expression (**Li et al., 2008**; **Zhang et al., 2018**; **Zhu et al., 2015**); NKX6-3, a gastrointestinal transcription factor (**Alanentalo et al., 2006**); and GIP, a gastrointestinal peptide involved in intestinal motility and energy homeostasis (**Adriaenssens et al., 2019**; **Jörnvall et al., 1981**; **Moody et al., 1984**; **Pederson and McIntosh, 2016**). Several less frequent (but still shared) novel antigens were also chosen for validation, including ASMT, a pineal gland enzyme involved in melatonin synthesis (**Ackermann et al., 2006**; **Rath et al., 2016**); and PDX1, an intestinal and pancreatic transcription factor (**Holland et al., 2002**; **Stoffers et al., 1997**; **Figure 3A**). Of note, this group of seven novel antigens all exhibited either tissue enriched, tissue enhanced, or group enhanced expression according to the Human Protein Atlas database (https://www.proteinatlas.org/about/download; **Uhlén et al., 2015**; **Supplementary file 3**). Using a whole-protein radiolabeled binding assay (RLBA) for validation, all seven proteins were immunoprecipitated by antibodies in both the PhIP-Seq APS1 discovery cohort (n = 39), as well as in the validation cohort of APS1 sera that had not been interrogated by PhIP-Seq (n = 28). Whereas an expanded set of non-APS1 controls (n = 61) produced little to no immunoprecipitation signal by RLBA as compared to positive control antibodies (low antibody index), APS1 samples yielded significant immunoprecipitation signal enrichment for each whole protein assay (high antibody index) (**Figure 3B** and **Supplementary file 4**).

The comparison of PhIP-Seq data to the results from the RLBAs (n = 39, discovery cohort only) yielded positive correlations between the two datasets (r = 0.62–0.95; **Figure 3—figure supplement 1**). Notably, for some antigens, such as NLRP5, and particularly for ASMT, the RLBA results revealed additional autoantibody-positive samples not detected by PhIP-Seq (**Figure 3B** and **Figure 3—figure supplement 1** and **Figure 1—figure supplement 2**).

## Autoantibody-disease associations for both known and novel antigens

Because the individuals in this APS1 cohort have been extensively phenotyped for 24 clinical manifestations, the PhIP-Seq APS1 data was queried for phenotypic associations. Several autoantibody specificities, both known and novel, were found to possess highly significant associations with several clinical phenotypes (*Figure 4* and *Figure 4—figure supplement 1*). Among these were the associations of KHDC3L with ovarian insufficiency, RFX6 with diarrheal-type intestinal dysfunction, CYP11A1 (also known as cholesterol side chain cleavage enzyme) with adrenal insufficiency (AI), and SOX10 with vitiligo (*Figure 4*). Strikingly, anti-CYP11A1 antibodies are present in AI and are known to predict disease development (*Betterle et al., 2002*; *Petra et al., 2000*; *Winqvist et al., 1993*). Similarly, antibodies to SOX10, a transcription factor involved in melanocyte differentiation and

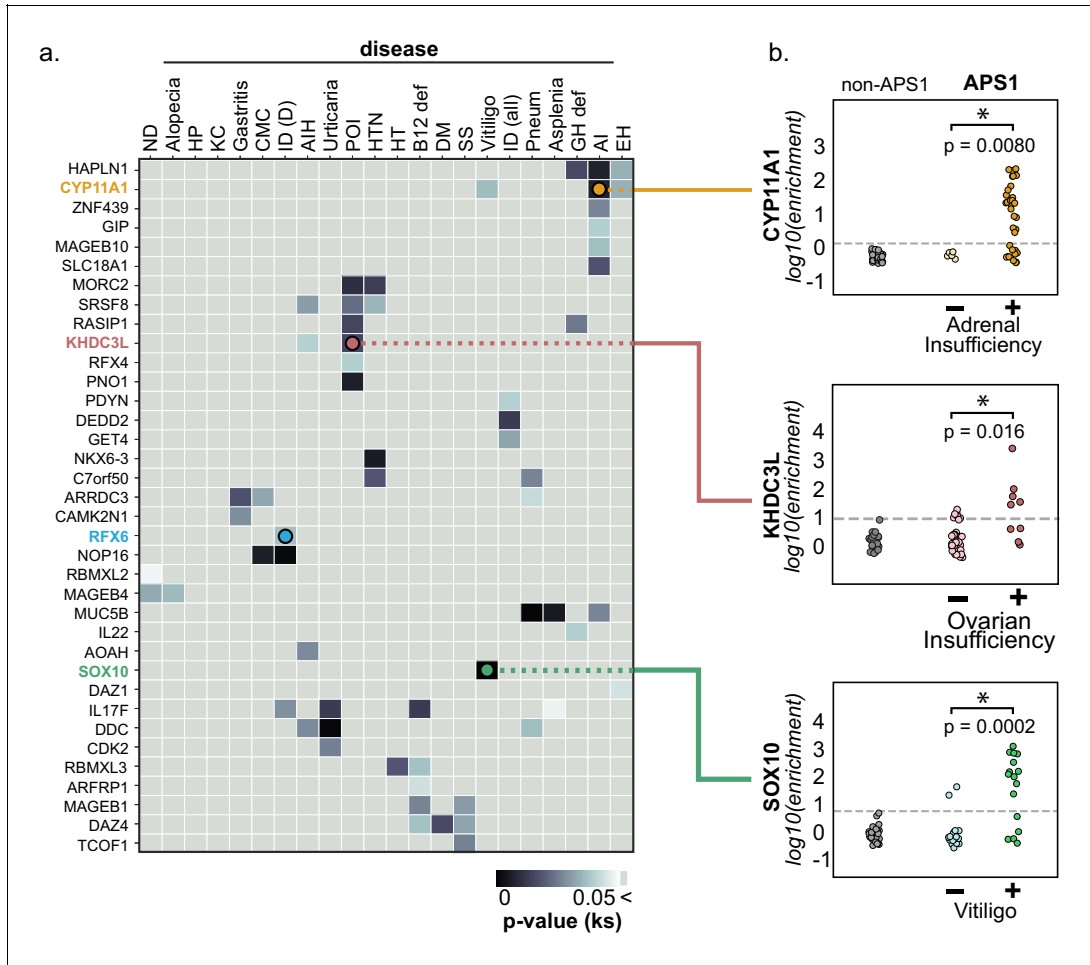

**Figure 4.** PhIP-Seq reproduces known clinical associations with anti-CYP11A1 and anti-SOX10 antibodies. (A) Heatmap of p-values (Kolmogorov-Smirnov testing) for differences in gene enrichments for individuals with versus without each clinical phenotype. Significant p-values in the negative direction (where mean PhIP-Seq enrichment is higher in individuals without disease) are masked (colored as p>0.05). (B) Anti-CYP11A1 PhIP-Seq enrichments are significantly different between APS1 patients with and without adrenal insufficiency (top panel; Kolmogorov-Smirnov test). Anti-SOX10 PhIP-Seq enrichments are significantly different between APS1 patients with and without Vitiligo (bottom panel). Anti-KHDC3L PhIP-Seq enrichments are significantly different between APS1 patients with and without ovarian insufficiency (middle panel). ND, nail dystrophy. HP, hypoparathyroidism. KC, keratoconjunctivitis. CMC, chronic mucocutaneous candidiasis. ID (D), Intestinal dysfunction (diarrheal-type). AIH, autoimmune hepatitis. POI, primary ovarian insufficiency. HTN, hypertension. HT, hypothyroidism. B12 def, B12 (vitamin) deficiency. DM, diabetes mellitus. SS, Sjogren's-like syndrome. Pneum, Pneumonitis. GH def, Growth hormone deficiency. AI, Adrenal Insufficiency. EH, (dental) enamel hypoplasia.

The online version of this article includes the following figure supplement(s) for figure 4:

**Figure supplement 1.** Clustered disease correlations in the APS1 cohort (Spearman's rank correlation; n = 67).
**Figure supplement 2.** KHDC3L is highly expressed in oocytes (top), but not in granulosa cells (bottom).

maintenance, have been previously shown to correlate with the presence of autoimmune vitiligo (*Hedstrand et al., 2001*).

## Anti-KHDC3L antibodies in APS1-associated ovarian insufficiency

Primary ovarian insufficiency is a highly penetrant phenotype, with an estimated 60% of females with APS1 progressing to an early, menopause-like state (*Ahonen et al., 1990*; *Ferre et al., 2016*). Interestingly, a set of 5 proteins (KHDC3L, SRSF8, PNO1, RASIP1, and MORC2) exhibited a significant association with ovarian insufficiency in this cohort (*Figure 4*). A publicly available RNA-sequencing dataset from human oocytes and supporting granulosa cells of the ovary confirmed that of these 5 genes, only *KHDC3L* exhibited expression levels in female oocytes comparable to the expression levels seen for the known oocyte markers *NLRP5* and *DDX4* (*Zhang et al., 2018*; *Figure 4—figure supplement 2*). We therefore chose to further investigate the relationship between anti-KHDC3L antibodies and ovarian insufficiency in our cohort (*Figure 5*).

KHDC3L is a well-studied molecular binding partner of NLRP5 within the ovary (*Li et al., 2008*; *Zhu et al., 2015*). Together, NLRP5 and KHDC3L form part of a critical oocyte-specific molecular complex, termed the subcortical maternal complex (SCMC) (*Bebbere et al., 2016*; *Li et al., 2008*; *Liu et al., 2016*; *Zhu et al., 2015*). Furthermore, knockout of the *NLRP5* and *KHDC3L* in female mice results in fertility defects, and human genetic mutations in these genes of the SCMC have been linked to infertility and molar pregnancies (*Akoury et al., 2015*; *Li et al., 2008*; *Reddy et al., 2013*;

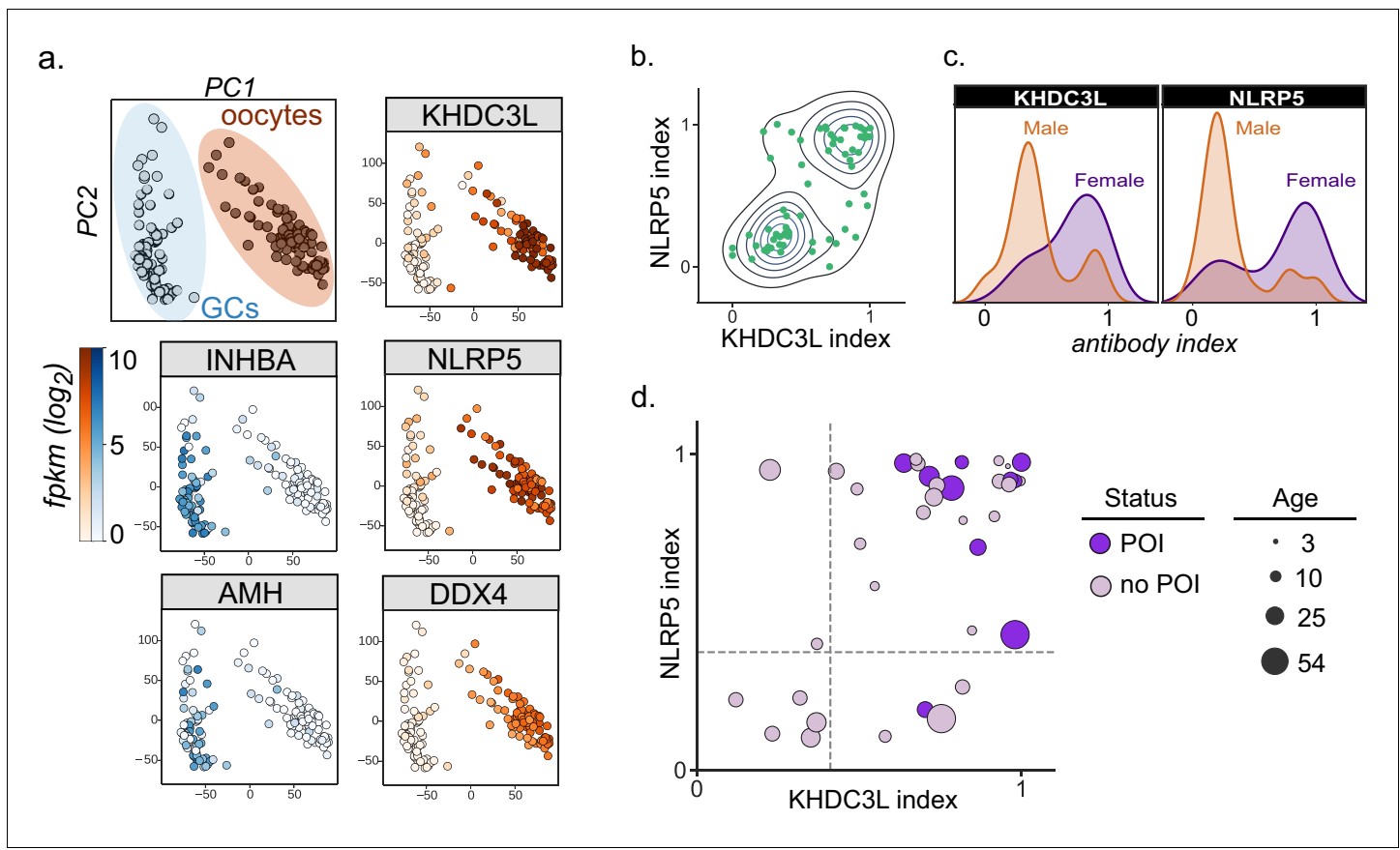

**Figure 5.** Autoantibodies to oocyte-expressed protein KHDC3L are associated with ovarian insufficiency. (A) Principle component analysis of transcriptome of single human oocytes (red) and granulosa cells (GCs, blue); data re-analyzed from *Zhang et al. (2018)*. KHDC3L is highly expressed in oocytes, along with binding partner NLRP5 and known oocyte marker DDX4. For comparison, known GC markers INHBA and AMH are primarily expressed in the GC population. (B) APS1 sera that are positive for one of anti-KHDC3L and anti-NLRP5 autoantibodies tend to also be positive for the other. (C) Antibody indices for both KHDC3L and NLRP5 are increased in females with APS1. (D) Antibody indices for females with APS1 by age; All 10 patients with primary ovarian insufficiency (POI) are positive for anti-KHDC3L antibodies. Of note, many of the individuals with anti-KHDC3L antibodies but without POI are younger and therefore cannot be fully evaluated for ovarian insufficiency.

*Wang et al., 2018*; *Zhang et al., 2019*). Interestingly, previous work established NLRP5 as a parathyroid-specific antigen in APS1, with potential for additional correlation with ovarian insufficiency (*Alimohammadi et al., 2008*; *Brozzetti et al., 2015*). However, anti-NLRP5 antibodies lack sensitivity for ovarian insufficiency. Importantly, unlike NLRP5, KHDC3L is expressed primarily in the ovary, and thus potentially represents a more oocyte-specific autoantigen (*Liu et al., 2016*; *Virant-Klun et al., 2016*; *Zhang et al., 2018*). Using the dataset from Zhang et. al, we confirmed that KHDC3L, as well as NLRP5 and the known oocyte marker DDX4, are highly expressed within the oocyte population, but not in the supporting granulosa cell types (*Zhang et al., 2018*; *Figure 5A*). Interestingly, the majority (64%) of APS1 sera had a concordant status for antibodies to KHDC3L and NLRP5 (*Figure 5B*). Although previous reports did not find a strong gender prevalence within samples positive for anti-NLRP5 antibodies, the mean anti-NLRP5 and anti-KHDC3L antibody signals were increased in females in this cohort (by 1.6- and 2.1-fold, respectively; *Figure 5C*). Finally, all 10 females in the expanded APS1 cohort with diagnosed ovarian insufficiency were also positive for anti-KHDC3L antibodies (*Figure 5D*).

## High prevalence of anti-ACP4 antibodies

Similar to known antigens CYP11A1, SOX10, and LCN1, the novel antigen ACP4 was found to occur at high frequencies in this cohort (*Figure 3A*). ACP4 (acid phosphatase 4) is highly expressed in dental enamel, and familial mutations in the *ACP4* gene result in dental enamel hypoplasia similar to the enamel hypoplasia seen in ~90% of this APS1 cohort (*Seymen et al., 2016*; *Smith et al., 2017*). Strikingly, 50% of samples were positive for anti-ACP4 antibodies by RLBA, with excellent correlation between RLBA and PhIP-Seq data (*Figure 3B* and *Figure 3—figure supplement 1A*). Consistently, samples from individuals with enamel hypoplasia exhibited a trend towards higher anti-ACP4 antibody signal by RLBA (*Figure 3—figure supplement 1B*, p=0.064).

## High prevalence of anti-RFX6 antibodies

In this cohort, 82% (55/67) of APS1 sera exhibited an RFX6 signal that was at least 3 standard deviations above the mean of non-APS1 control signal due to the extremely low RFX6 signal across all non-APS1 controls by RLBA (*Figure 3B*). Using a more stringent cutoff for RFX6 positivity by RLBA at 6 standard deviations above the mean, 65% of APS1 samples were positive for anti-RFX6 antibodies. RFX6 is expressed in both intestine and pancreas, and loss of function RFX6 variants in humans lead to both intestinal and pancreatic pathology (*Gehart et al., 2019*; *Patel et al., 2017*; *Piccand et al., 2019*; *Smith et al., 2010*). Interestingly, across all samples with anti-RFX6 antibodies, the response targeted multiple sites within the protein, suggesting a polyclonal antibody response (*Figure 6—figure supplement 1A*).

## Anti-enteroendocrine and anti-RFX6 response in APS1

The extent and frequency of intestinal dysfunction in people with APS1 has only recently been clinically uncovered and reported, and therefore still lacks unifying diagnostic markers as well as specific intestinal target antigen identities (*Ferre et al., 2016*). This investigation of APS1 sera revealed several antigens that are expressed in the intestine, including RFX6, GIP, PDX1, and NKX6-3. We chose to further study whether autoimmune response to RFX6+ cells in the intestine was involved in APS1-associated intestinal dysfunction. Using a publicly available murine single-cell RNA sequencing dataset of 16 different organs and over 120 different cell types, *RFX6* expression was confirmed to be present in and restricted to pancreatic islets and intestinal enteroendocrine cells (*Schaum et al., 2018*; *Figure 6—figure supplement 1B and C*). Serum from an individual with APS1-associated intestinal dysfunction and anti-RFX6 antibodies was next tested for reactivity against human intestinal enteroendocrine cells, revealing strong nuclear staining that colocalized with ChromograninA (ChgA), a well-characterized marker of intestinal enteroendocrine cells (*Goldspink et al., 2018*; *O'Connor et al., 1983*; *Figure 6A*, right panel and inset). In contrast, enteroendocrine cell staining was not observed from APS1 samples that lacked anti-RFX6 antibodies or from non-APS1 control samples. (*Figure 6A*, center and left panels). Furthermore, serum from samples with anti-RFX6 antibodies stained transfected tissue culture cells expressing RFX6 (*Figure 6B*, *Figure 6—figure supplement 2*). These data support the notion that there exists a specific antibody signature, typified by anti-RFX6 antibodies, associated with enteroendocrine cells in APS1.

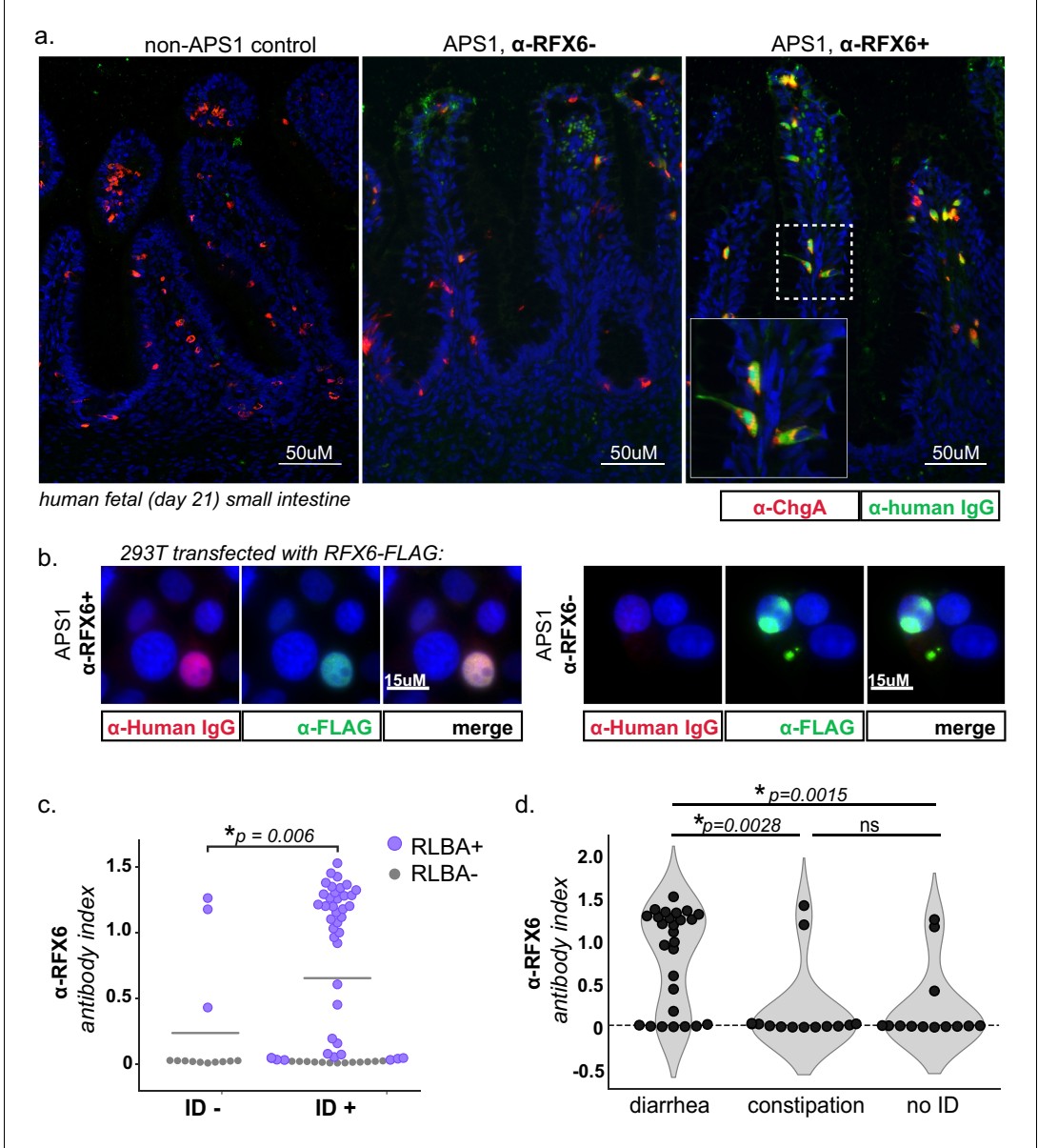

**Figure 6.** APS1 patients with intestinal dysfunction mount an antibody response to intestinal enteroendocrine cells and to enteroendocrine-expressed protein RFX6. (**A**) Anti-RFX6 positive APS1 serum with intestinal dysfunction co-stains Chromogranin-A (ChgA) positive enteroendocrine cells in a nuclear pattern (right panel and inset). In contrast, non-APS1 control sera as well as anti-RFX6 negative APS1 serum do not co-stain ChgA+ enteroendocrine cells (left and center panels). (**B**) Anti-RFX6+ serum, but not anti-RFX6- serum, co-stains HEK293T cells transfected with an RFX6-expressing plasmid (see also: *Figure 6—figure supplement 2*). (**C**) Radioligand binding assay (RLBA) anti-RFX6 antibody index is significantly higher across individuals with intestinal dysfunction (ID; Mann-Whitney U, p=0.006). Purple color indicates samples that fall above 6 standard deviations of the mean non-APS1 control RLBA antibody index. (**D**) Individuals with the diarrheal subtype of ID have a higher frequency of anti-RFX6 antibody positivity as compared to those with constipation-type ID (Mann-Whitney U, p=0.0028) or no ID (p=0.0015).

The online version of this article includes the following figure supplement(s) for figure 6:

**Figure supplement 1.** Higher resolution PhIP-seq and transcriptional data for novel autoantigen RFX6.

**Figure supplement 2.** Anti-RFX6+ sera (top two panels), but not anti-RFX6- serum or non-APS1 control serum (bottom two panels), co-stain HEK293T cells transfected with an RFX6-expressing plasmid.

**Figure supplement 3.** Extended validation and clinical correlations for intestinal antigens RFX6 and TPH1.

Both mice and humans with biallelic mutation of the gene encoding RFX6 have enteroendocrine cell deficiency and intestinal malabsorption (*Mitchell et al., 2004*; *Piccand et al., 2019*; *Smith et al., 2010*), and humans with other forms of genetic or acquired enteroendocrine cell deficiency also suffer from chronic malabsorptive diarrhea (*Högenauer et al., 2001*; *Mitchell et al., 2004*; *Oliva-Hemker et al., 2006*; *Posovszky et al., 2012*; *Wang et al., 2006*). In this cohort, 54/67 (81%) of individuals have intestinal dysfunction defined as the presence of chronic diarrhea, chronic constipation or an alternating pattern of both, without meeting ROME III diagnostic criteria for irritable bowel syndrome, as previously described (*Ferre et al., 2016*). When the cohort was subsetted by presence or absence of intestinal dysfunction, the anti-RFX6 RLBA signal was significantly higher when intestinal dysfunction was present (*Figure 6C*). Further subsetting of the cohort by subtype of intestinal dysfunction revealed that individuals with anti-RFX6 antibodies belonged preferentially to the diarrheal-type (as opposed to constipation-type) group of intestinal dysfunction (*Figure 6D* and *Figure 6—figure supplement 3A*). In contrast, antibodies to the known intestinal antigen TPH1 were distributed less specifically across both types (diarrheal- and constipation-type) of ID, despite a high frequency of both anti-TPH1 and anti-RFX6 in the cohort (55/67 for anti-RFX6, 45/67 for anti-TPH1) (*Figure 6—figure supplement 3C and D*). These observations are consistent with a previous report that anti-TPH1 antibodies show an association with ID, but not specifically with the diarrheal subtype (*Kluger et al., 2015*).

Given that RFX6 is also expressed in the pancreas, we also examined the association of anti-RFX6 antibodies with APS1-associated type 1 diabetes. We observed that 6/7 APS1-associated type 1 diabetes samples had positive signal for anti-RFX6 antibodies by RLBA (*Figure 6—figure supplement 3B*). However, due to small sample size, an expanded cohort would be needed to determine the significance of this observation. Together, these data suggest that RFX6 is a common, shared autoantigen in APS1 that may be involved in the immune response to intestinal enteroendocrine cells as well as pancreatic islets. Future studies will help to determine whether testing for anti-RFX6 antibodies possesses clinical utility for prediction or diagnosis of specific APS1 autoimmune disease manifestations as well as for non-APS1 autoimmune disease.

## Discussion

Here, we have identified a new set of autoantigens that are associated with autoimmune features in APS1 by using the broad-based antigen screening platform of PhIP-Seq. Unlike fixed protein arrays, programmable phage display possesses the advantage of being able to comprehensively cover all annotated proteins and their isoforms. The PhIP-Seq library used here is composed of over 700,000 peptides, each 49 amino acids, and corresponding to approximately 20,000 proteins and their known splicing isoforms. This is highly complementary to recently published protein arrays that cover approximately 9000 distinct proteins (*Fishman et al., 2017*; *Landegren et al., 2016*; *Meyer et al., 2016*). Recent protein array approaches with APS1 samples using strict cutoffs have been able to identify a number of new autoantigen targets that include PDILT and MAGEB2 (*Landegren et al., 2016*). Several new targets, including RFX6, KHDC3L, ACP4, NKX6-3, ASMT, and PDX1, were likely discovered here because these antigens were not present on previously published protein array platforms. Only a subset of the novel targets identified here were validated orthogonally. While none failed validation relative to non-APS1 controls, further validation work will be needed for the many additional novel targets identified by PhIP-Seq. It is also worth mentioning that the PhIP-Seq method leverages continuing declines in the cost of oligonucleotide synthesis and Next-Generation Sequencing. Both technologies benefit from economies of scale, and once constructed, a PhIP-Seq phage library may be propagated in large quantities at negligible cost. The primary disadvantage of PhIP-Seq is the fact that conformation specific antibodies are likely to be missed, unless short linear subsequences carry significant binding energy. For example, PhIP-Seq detected only limited signal towards some literature reported antigens, including GAD65 and interferon family proteins in this APS1 cohort. Given that these antigens have been reported to involve conformational epitopes, antibodies to these antigens would not be predicted to be easily detected by linear peptides (*Björk et al., 1994*; *Meager et al., 2006*; *Meyer et al., 2016*; *Wolff et al., 2013*; *Ziegler et al., 1996*). Nonetheless, the ability to detect anti-interferon antibodies in a subset of APS1 samples highlights the utility of PhIP-Seq for antigen discovery despite decreased sensitivity for certain epitopes (*Figure 1—figure supplement 1*).

People with (*Anderson et al., 2002*; *Cheng and Anderson, 2018*; *Husebye et al., 2018*; *Malchow et al., 2016*) APS1 develop autoimmune manifestations over the course of many years, and it is thought that each manifestation may be explained by autoimmune response to one or few initial protein targets. In principle, these target proteins would most likely (1) exhibit thymic AIRE-dependency and (2) be restricted to the single or narrow range of tissues associated with the corresponding autoimmune disease. For example, adrenal insufficiency, which results from autoimmune response to cells of the adrenal gland, is thought to occur due to targeting of adrenally-expressed cytochrome p450 family members (*Petra et al., 2000*; *Winqvist et al., 1993*). However, a more complete understanding of the protein target spectrum paired with clinical phenotypic associations has been lacking. This, combined with the limited applicability of murine observations to the human disease, has left the question of which clinical characteristics best associate with APS1 autoantigens a heavily debated subject (*Pöntynen et al., 2006*).

Testing for defined autoantibody specificities provides substantial clinical benefit for prediction and diagnosis of autoimmune disease. A primary goal of this study was to identify autoantigens with potential clinical significance; consistently, our analyses focused primarily on antigens that appeared across multiple samples, rather than autoantigens that were restricted to individual samples. Using conservative inclusion criteria, we discovered 69 novel autoantigens that were shared across a minimum of 3 APS1 samples, of which 7/7 were successfully validated at the whole protein level. Overall, we have expanded the known repertoire of common APS1 antigens, confirming that the antibody target repertoire of common antigens in APS1 is larger than previously appreciated. Interestingly, our data also suggest that the size of the commonly autoantibody-targeted repertoire of proteins is dramatically lower than the number of genes (~4000) that exhibit AIRE-dependent thymic expression.

The spectrum of different autoimmune diseases that can be observed in APS1 is extensive and has continued to expand through investigation of larger cohorts (*Ahonen et al., 1990*; *Bruserud et al., 2016*; *Ferre et al., 2016*). In this study, clinical metadata encompassing disease status across 24 individual disease manifestations in a total of 67 people with APS1 was leveraged to uncover (among others) an association of anti-KHDC3L antibodies and ovarian insufficiency, a disease that affects over half of all women with APS1 and manifests as abnormal menstrual cycling, reduced fertility, and early menopause. While autoreactivity to the steroidogenic granulosa cells – the cells surrounding and supporting the oocytes – has been proposed as one etiology of the clinical ovarian insufficiency, it has also been suggested that there may exist an autoimmune response to the oocyte itself (*Jasti et al., 2012*; *Maclaren et al., 2001*; *Otsuka et al., 2011*; *Petra et al., 2000*; *Welt, 2008*). Our finding that females with APS1-associated ovarian insufficiency exhibit autoantibodies to KHDC3L, an oocyte specific protein, supports this hypothesis. As exemplified by autoantibody presence in other autoimmune conditions, anti-KHDC3L antibodies may also have predictive value. Specifically, in our cohort, we found anti-KHDC3L antibodies to be present in many young, pre-menstrual females; these observations will require additional studies in prospective, longitudinal cohorts for further evaluation of potential predictive value. Interestingly, primary ovarian insufficiency (POI) in the absence of AIRE-deficiency is increasingly common and affects an estimated 1 in 100 women; up to half of these cases have been proposed to have autoimmune etiology (*Huhtaniemi et al., 2018*; *Jasti et al., 2012*; *Nelson, 2009*; *Silva et al., 2014*). The detection of anti-NLRP5 antibodies in a small subset of patients with non-APS1 ovarian insufficiency in a previous study further underlines the importance of evaluating the prevalence of anti-KHDC3L antibodies in women with POI (*Brozzetti et al., 2015*).

We noted that the majority of samples with antibodies to KHDC3L also exhibited antibodies to NLRP5, and vice versa. Remarkably, both of these proteins are critical parts of a subcortical maternal complex (SCMC) in both human and murine oocytes (*Li et al., 2008*; *Zhu et al., 2015*). Indeed, 'multi-pronged' targeting of the same pathway has been previously implicated in APS1, where antibodies to DDC and TPH1 – enzymes in the serotonin and melatonin synthesis pathways – have been described (*Ekwall et al., 1998*; *Husebye et al., 1997*; *Kluger et al., 2015*; *Rorsman et al., 1995*). In addition to these targets, our data revealed an additional autoantibody-targeted enzyme ASMT in the same melatonin synthesis pathway. While the earlier TPH1- and DDC-catalyzed steps occur in both the intestine and pineal gland and precede the formation of serotonin, ASMT is predominantly expressed in the pineal gland and catalyzes the last, post-serotonin step in melatonin synthesis,

suggesting that targeting of this pathway occurs at multiple distinct steps. To our knowledge, this is the first reported autoantigen in APS1 whose expression is restricted to the central nervous system.

In past and ongoing investigations, some individuals with APS1 have been reported to feature histologic loss of intestinal enteroendocrine cells on biopsy (*Högenauer et al., 2001*; *Oliva-Hemker et al., 2006*; *Posovszky et al., 2012*; Natarajan et al., manuscript in preparation). The association of anti-RFX6 antibodies with the diarrheal type of intestinal dysfunction is consistent with published studies in murine models of *Rfx6* (and enteroendocrine cell) ablation (*Piccand et al., 2019*; *Smith et al., 2010*). In addition, human enteroendocrine cell deficiency as well as mutations in enteroendocrine gene *NEUROG3* have been linked to chronic diarrhea and malabsorption, and recently, intestinal enteroendocrine cells have been suggested to play a role in mediating intestinal immune tolerance (*Ohsie et al., 2009*; *Sifuentes-Dominguez et al., 2019*; *Wang et al., 2006*). In sum, although APS1-associated intestinal dysfunction may have multiple etiologies, including autoimmune enteritis or dysfunction of exocrine pancreas, our findings of highly prevalent anti-RFX6 antibodies provide evidence of a common, shared autoantigen involved with this disease phenotype. In addition, patients with type 1 diabetes alone (not in association with APS1) frequently exhibit intestinal dysfunction related to multiple etiologies including Celiac disease, autonomic neuropathy, and exocrine pancreatic insufficiency (*Du et al., 2018*); future studies will be needed to determine whether anti-RFX6 antibodies may distinguish a subset of these patients with an autoimmune enteroendocrinopathy contributing to their symptoms.

While we report many novel antigens, we also acknowledge that the relationship between autoantibody status and disease is often complicated. This concept can be illustrated by examining the well-established autoantibody specificities in autoimmune diabetes (*Taplin and Barker, 2008*). First, islet autoantibodies (GAD65, ZNT8, etc.) can be found within non-autoimmune sera, where they are thought to represent an increased risk of developing disease as compared to the antibody-negative population. Second, not all patients with autoimmune diabetes are autoantibody positive. In sum, while autoantibodies can be extremely useful for risk assessment as well as for diagnosis, they often lack high sensitivity and specificity; both of these caveats can result in difficulties detecting strong clinical associations. For example, anti-ACP4 antibodies are highly prevalent in our cohort, but they exhibit only a trending association with dental enamel hypoplasia despite the strong biological evidence that ACP4 dysfunction leads to enamel hypoplasia (*Seymen et al., 2016*; *Smith et al., 2017*). Our data in humans is currently insufficient to determine whether immune responses to novel antigens such as ACP4 are pathogenic, indirectly linked to risk of disease, or instead simply represent a B-cell bystander effect. To better address these questions, we propose that future studies in mouse models could elucidate whether immune response to specific proteins, including ACP4, can result in the proposed phenotypes.

As the spectrum of diseases with potential autoimmune etiology continues to expand, the characteristic multiorgan autoimmunity in APS1 provides an ideal model system to more broadly approach the question of which proteins and cell types tend to be aberrantly targeted by the immune system. The data presented here has illuminated a collection of novel human APS1 autoimmune targets, as well as a novel antibody-disease association between RFX6 and diarrheal-type intestinal dysfunction, a highly prevalent disorder in APS1 that has until now lacked clinically applicable predictive or diagnostic markers. In sum, these data have significantly expanded the known autoantigen target profile in APS1 and highlighted several new directions for exploring the mechanics and clinical consequences of this complex syndrome.

# Materials and methods

**Key resources table**

| Reagent type (species) or resource | Designation | Source or reference | Identifiers | Additional information |
|---|---|---|---|---|
| Transfected construct (human) | 293T mock transfection construct | Origene | PS100001 | |

*Continued on next page*

*Continued*

| Reagent type (species) or resource | Designation | Source or reference | Identifiers | Additional information |
|---|---|---|---|---|
| Transfected construct (human) | 293T RFX6 transfection construct | Origene | RC206174 | |
| Antibody | Goat Anti-NLRP5, polyclonal | Santa Cruz | sc-50630 | RLBA 1:50 |
| Antibody | Mouse Anti-SOX10, monoclonal | Abcam | ab181466 | RLBA 1:25 |
| Antibody | Sheep Anti-RFX6, polyclonal | R and D Systems | AF7780 | RLBA 1:50 |
| Antibody | Rabbit Anti-KHDC3L, polyclonal | Abcam | ab170298 | RLBA 1:25 |
| Antibody | Rabbit Anti-CYP11A1, polyclonal | Abcam | ab175408 | RLBA 1:50 |
| Antibody | Rabbit Anti-NKX6-3, polyclonal | Biorbyt | orb127108 | RLBA 1:50 |
| Antibody | Mouse Anti-GIP, monoclonal | Abcam | ab30679 | RLBA 1:50 |
| Antibody | Rabbit Anti-PDX1, polyclonal | Invitrogen | PA5-78024 | RLBA 1:50 |
| Antibody | Rabbit Anti-ASMT, polyclonal | Invitrogen | PA5-24721 | RLBA 1:25 |
| Antibody | Rabbit Anti-CHGA, polyclonal | Abcam | ab30679 | IF 1:5000 |
| Antibody | Rabbit Anti-DYKDDDDK (D6W5B), monoclonal | Cell Signaling Technologies | #14793 | RLBA 1:125, IF 1:2000 |

## Data collection

All patient cohort data was collected and evaluated at the NIH, and all APECED/APS1 patients were enrolled in a research study protocols approved by the NIAID, NIH Clinical Center, and NCI Institutional Review Board Committee and provided with written informed consent for study participation. All NIH patients gave consent for passive use of their medical record for research purposes (protocol #11-I-0187). The majority of this cohort data was previously published by *Ferre et al. (2016)* and *Ferré et al. (2019)*.

## Phage immunoprecipitation – Sequencing (PhIP-Seq)

For PhIP-Seq, we adapted a custom-designed phage library consisting of 731,724 49AA peptides tiling the full protein-coding human genome including all isoforms (as of 2016) with 25AA overlap as previously described (*O'Donovan et al., 2018*). 1 milliliter of phage library was incubated with 1 microliter of human serum overnight at 4C, and human antibody (bound to phage) was immunoprecipitated using 40 ul of a 1:1 mix of protein A/G magnetic beads (Thermo Fisher, Waltham, MA, #10008D and #10009D). Beads were washed 4 times and antibody-bound phage were eluted into 1 ml of *E. coli* at OD of 0.5–0.7 (BLT5403, EMD Millipore, Burlington, MA) for selective amplification of eluted phage. This library was re-incubated with human serum and repeated, followed by phenol-chloroform extraction of DNA from the final phage library. DNA was barcoded and amplified (Phusion PCR, 30 rounds), gel purified, and subjected to Next-Generation Sequencing on an Illumina MiSeq Instrument (Illumina, San Diego, CA).

## PhIP-Seq analysis

Sequencing reads from fastq files were aligned to the reference oligonucleotide library and peptide counts were subsequently normalized by converting raw reads to percentage of total reads per

sample. Peptide and gene-level enrichments for both APS1 and non-APS1 sera were calculated by determining the fold-change of read percentage per peptide and gene in each sample over the mean read percentage per peptide and gene in a background of mock-IP (A/G bead only, n = 17). Individual samples were considered 'positive' for genes where the enrichment value was 10-fold or greater as compared to mock-IP. For plotting of multiple genes in parallel (*Figures 1* and *2*), enrichment values were z-scored and hierarchically clustered using Pearson correlation.

## Statistics

For comparison of distribution of PhIP-Seq gene enrichment between APS1 patients with and without specific disease manifestations, a (non-parametric) Kolmogorov-Smirnov test was used.

For radioligand binding assays, antibody index for each sample was calculated as follows: (sample value – mean blank value) / (positive control antibody value – mean blank value). Comparison of antibody index values between non-APS1 control samples and APS1 samples was performed using a Mann-Whitney *U* test. Experimental samples that fell 3 standard deviations above of the mean of non-APS1 controls for each assay were considered positive, except in the case of RFX6, where a cutoff of 6 standard deviations above the mean of non-APS1 controls was used.

## Assessing tissue-specific RNA expression

To determine tissue-specificity and tissue-restriction of *Rfx6* expression in mice, we used publicly available Tabula Muris data (tabula-muris.ds.czbiohub.org) (*Schaum et al., 2018*). For investigation of *KHDC3L* expression in human ovary, we downloaded publicly available normalized FPKM transcriptome data from human oocytes and granulosa cells (GSE107746_Folliculogenesis_FPKM.log2. txt) (*Zhang et al., 2018*). With this data, we performed principle component analysis, which clustered the two cell types correctly according to their corresponding sample label, and plotted log2 (FPKM) by color for each sample.

## 293T overexpression assays

Human kidney embryo 293T (ATCC, Manassas, VA, #CRL-3216, RRID:CVCL_0063) cells were plated at 30% density in a standard 24-well glass bottom plate in complete DMEM media (Thermo Fisher, #119651198) with 10% Fetal Bovine Serum (Thermo Fisher, #10438026), 292 ug/ml L-glutamine, 100 ug/ml Streptomycin Sulfate, and 120Units/ml of Penicillin G Sodium (Thermo Fisher, #10378016). 18 hr later, cells were transiently transfected using a standard calcium chloride transfection protocol. For transfections, 0.1 ug of sequence-verified pCMV-insert-MYC-FLAG overexpression vectors containing either no insert (Origene #PS100001; 'mock' transfection) or RFX6 insert (Origene #RC206174) were transfected into each well. 24 hr post-transfection, cells were washed in 1X PBS and fixed in 4% PFA for 10 min at room temperature. 293T cells were tested as negative for mycoplasma contamination.

## 293T indirect immunofluorescence

Fixed 293T cells were blocked for 1 hr at room temperature in 5% BSA in PBST. For primary antibody incubation, cells were incubated with human serum (1:1000) and rabbit anti-FLAG antibody (1:2000) in 5% BSA in PBST for 2 hr at room temperature (RT). Cells were washed 4X in PBST and subsequently incubated with secondary antibodies (goat anti-rabbit IgG 488, Invitrogen, Carlsbad, CA; #A-11034, 1:4000; and goat anti-human 647, Invitrogen #A-21445, 1:4000) for 1 hr at room temperature. Finally, cells were washed 4X in PBST, incubated with DAPI for 5 min at RT, and subsequently placed into PBS for immediate imaging. All images were acquired with a Nikon Ti inverted fluorescence microscope (Nikon Instruments, Melville, NY).

## Indirect dual immunofluorescence on human fetal intestine

Human fetal small bowels (21.2 days gestational age) were processed as previously described (*Berger et al., 2015*). Individual APS1 sera (1:4000 dilution) were used in combination with rabbit antibodies to human Chromogranin A (Abcam, Cambridge, MA; #ab15160, 1:5000 dilution). Immunofluorescence detection utilized secondary Alexa Fluor secondary antibodies (Life Technologies, Waltham, MA; 488 goat anti-human IgG, #A11013, RRID:AB_141360; and 546 goat anti-rabbit IgG, #A11010, RRID:AB_2534077). Nuclear DNA was stained with Hoechst dye (Invitrogen, #33342). All

images were acquired with a Leica SP5 White Light confocal laser microscope (Leica Microsystems, Buffalo Grove, IL).

## 35S-radiolabeled protein generation and binding assay

DNA plasmids containing full-length cDNA under the control of a T7 promoter for each of the validated antigens (*Supplementary file 4*) were verified by Sanger sequencing and used as DNA templates in the T7 TNT in vitro transcription/translation kit (Promega, Madison, WI; #L1170) using [35S]-methionine (PerkinElmer, Waltham, MA; #NEG709A). Protein was column-purified on Nap-5 columns (GE healthcare, Chicago, IL; #17-0853-01) and immunoprecipitated on Sephadex protein A/G beads (Sigma Aldrich, St. Louis, MO; #GE17-5280-02 and #GE17-0618-05, 4:1 ratio) in duplicate with serum or control antibodies in 96-well polyvinylidene difluoride filtration plates (Corning, Corning, NY; #EK-680860). Each well contained 35'000 counts per minute (cpm) of radiolabeled protein and 2.5 ul of serum or appropriately diluted control antibody (*Supplementary file 4*). The cpms of immunoprecipitated protein was quantified using a 96-well Microbeta Trilux liquid scintillation plate reader (Perkin Elmer).

## Acknowledgements

We thank Joseph M Replogle, Jeffrey A Hussmann, Madhura Raghavan, Hanna Retallack, and members of the DeRisi, Anderson, Lionakis, and German labs for helpful discussions. We thank Kari Herrington and the UCSF Nikon Imaging Center for imaging support, as well as Sabrina Mann, Wint Lwin, Lillian Khan, and the UCSF Center for Advanced Technology for technical support. We thank the New York Blood Bank for providing us with the de-identified human non-inflammatory control plasma samples used in this study. This work was supported by the Helmsley Foundation, the Parker Institute for Cancer Immunotherapy, the Chan Zuckerberg Biohub Initiative, the Larry L Hillblom Foundation, and the National Institute of Diabetes, Digestive and Kidney Disease, National Institute of Health, Bethesda, Maryland, USA. This work was also supported by the Division of Intramural Research (DIR) of the National Institute of Allergy and Infectious Diseases, National Institute of Health.

## Additional information

### Competing interests

Sara E Vazquez: JD, MSA, and SEV have a provisional patent on clinical application of autoantigens described in this study. Joseph L DeRisi: JD is a scientific advisory board member of Allen and Company. JD, MSA, and SEV have a provisional patent on clinical application of autoantigens described in this study. Mark S Anderson: MSA owns stock in Merck and Medtronic. JD, MSA, and SEV have a provisional patent on clinical application of autoantigens described in this study. The other authors declare that no competing interests exist.

### Funding

| Funder | Grant reference number | Author |
| --- | --- | --- |
| National Institute of Allergy and Infectious Diseases | 5P01AI118688-04 | Mark S Anderson |
| National Institute of Allergy and Infectious Diseases | 1ZIAAI001175-07 | Michail Lionakis |
| National Institute of Diabetes and Digestive and Kidney Diseases | 1F30DK123915-01 | Sara E Vazquez |
| Chan Zuckerberg Biohub | | Joseph L DeRisi |
| Larry L. Hillblom Foundation | | Michael German |
| Parker Institute for Cancer Immunotherapy | | Mark S Anderson |

| Juvenile Diabetes Research Foundation International | | Mark S Anderson |
|---|---|---|
| Helmsley Charitable Trust | | Mark S Anderson |
| National Institute of General Medical Sciences | 5T32GM007618-42 | Mark S Anderson |

The funders had no role in study design, data collection and interpretation, or the decision to submit the work for publication.

## Author contributions
Sara E Vazquez, Conceptualization, Data curation, Formal analysis, Funding acquisition, Validation, Investigation, Visualization, Methodology, Writing - original draft; Elise MN Ferré, Resources, Data curation, Investigation, Writing - review and editing; David W Scheel, Sara Sunshine, Validation, Investigation, Visualization, Writing - review and editing; Brenda Miao, Validation, Investigation; Caleigh Mandel-Brehm, Zoe Quandt, Alice Y Chan, Resources, Writing - review and editing; Mickie Cheng, Resources; Michael German, Resources, Supervision, Funding acquisition, Writing - review and editing; Michail Lionakis, Resources, Supervision, Funding acquisition, Project administration, Writing - review and editing; Joseph L DeRisi, Conceptualization, Resources, Software, Supervision, Funding acquisition, Visualization, Methodology, Writing - original draft, Project administration; Mark S Anderson, Conceptualization, Resources, Supervision, Funding acquisition, Writing - original draft, Project administration

## Author ORCIDs
Sara E Vazquez  https://orcid.org/0000-0002-0601-7001
Brenda Miao  http://orcid.org/0000-0002-3393-9837
Joseph L DeRisi  https://orcid.org/0000-0002-4611-9205
Mark S Anderson  https://orcid.org/0000-0002-3093-4758

## Ethics
Human subjects: All patient cohort data was collected and evaluated at the NIH, and all APECED/APS1 patients were enrolled in a research study protocols approved by the NIAID, NIH Clinical Center, and NCI Institutional Review Board Committee and provided with written informed consent for study participation. All NIH patients gave consent for passive use of their medical record for research purposes (protocol #11-I-0187). The majority of this human cohort data was previously published by Ferré et al. 2016 and Ferré et al. 2019.

## Decision letter and Author response
Decision letter https://doi.org/10.7554/eLife.55053.sa1
Author response https://doi.org/10.7554/eLife.55053.sa2

# Additional files

## Supplementary files
• Supplementary file 1. APS1 cohort: Clinical Data. ND, nail dystrophy. HP, hypoparathyroidism. KC, keratoconjunctivitis. CMC, chronic mucocutaneous candidiasis. ID (D, C, B), Intestinal dysfunction (diarrheal-type, constipation-type, both). AIH, autoimmune hepatitis. POI, primary ovarian insufficiency. HTN, hypertension. HT, hypothyroidism. B12 def, B12 (vitamin) deficiency. DM, diabetes mellitus. SS, Sjogren's-like syndrome. GH def, Growth hormone deficiency. AI, Adrenal Insufficiency. EH, (dental) enamel hypoplasia. TF, testicular failure. TIN, Tubulointerstitial Nephritis. Hpit, Hypopituitarism. UE, Urticarial eruption. D, Discovery cohort; V, Validation cohort. *Age at most recent evaluation

• Supplementary file 2. Non-APS1 control cohort: Clinical Data. D, Discovery cohort; V, Validation cohort.

- Supplementary file 3. Tissue-restricted expression patterns of validated and putative novel APS1 antigens.

- Supplementary file 4. Antibody information by application.

- Transparent reporting form

## Data availability

All sequencing data generated in this study are deposited on Dryad Digital Repository in conjunction with this submission (https://doi.org/10.7272/Q66H4FM2).

The following dataset was generated:

| Author(s) | Year | Dataset title | Dataset URL | Database and Identifier |
|---|---|---|---|---|
| Vazquez SE, Ferré EMN, Scheel DW, Sunshine S, Miao B, Mandel-Brehm C, Quandt Z, Chan AY, Cheng M, German MS, Lionakis MS, DeRisi JL, Anderson MS | 2020 | Data from: Identification of novel, clinically correlated autoantigens in the monogenic autoimmune syndrome APS1 by PhIP-Seq | https://doi.org/10.7272/Q66H4FM2 | Dryad Digital Repository, 10.7272/Q66H4FM2 |

The following previously published datasets were used:

| Author(s) | Year | Dataset title | Dataset URL | Database and Identifier |
|---|---|---|---|---|
| Zhang Y, Yan Z, Qin Q, Nisenblat V, Chang H-M, Yu Y, Wang T, Lu C, Yang M, Yang S, Yao Y, Zhu X, Xia X, Dang Y, Ren Y, Yuan P, Li R, Liu P, Guo H, Yan L | 2018 | Transcriptome Landscape of Human Folliculogenesis Reveals Oocyte and Granulosa Cell Interactions | https://www.ncbi.nlm.nih.gov/geo/query/acc.cgi?acc=GSE107746 | NCBI Gene Expression Omnibus, GSE107746 |
| Pisco AO | 2018 | Tabula Muris: Transcriptomic characterization of 20 organs and tissues from Mus musculus at single cell resolution | https://www.ncbi.nlm.nih.gov/geo/query/acc.cgi?acc=GSE109774 | NCBI Gene Expression Omnibus, GSE109774 |
| Human Protein Atlas | 2015 | Tissue-based map of the human proteome | https://www.proteinatlas.org/about/download | Human Protein Atlas, rna_tissue_consensus.tsv |

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
