## [Decision Letter]

**Acceptance summary:**

Vazquez and colleagues screen sera from patients with autoimmune polyglandular syndrome type 1 (APS1) for novel tissue-restricted autoantibodies using PhIP-seq. Using discovery and validation patient cohorts, they have identified multiple autoantibody-autoantigen pairs, including those previously demonstrated to be associated with APS1, and several novel ones, and verify them using an orthogonal method. They provide confirmation of tissue-restricted expression of the antigens and disease association of the discovered autoantibodies. The identification of novel autoantigens for Primary Ovarian Insufficiency (KHDC3L) and for intestinal dysfunction with an apparently distinct clinical phenotype (RFX-6) constitute major advances in the area and underlines the fact that autoantigens in this syndrome usually display tissue-specific expression.

**Decision letter after peer review:**

Thank you for submitting your article "Identification of novel, clinically correlated autoantigens in the monogenic autoimmune syndrome APS1 by PhIP-Seq" for consideration by *eLife*. Your article has been reviewed by two peer reviewers, and the evaluation has been overseen by a Reviewing Editor and Satyajit Rath as the Senior Editor. The following individual involved in review of your submission has agreed to reveal their identity: Joe Craft (Reviewer #1).

The reviewers have discussed the reviews with one another and the Reviewing Editor has drafted this decision to help you prepare a revised submission.

Summary:

The identification of novel autoantigens for Primary Ovarian Insufficiency (KHDC3L) and for intestinal dysfunction (RFX-6) constitute major advances in the area and underlines once again the fact that bona fide autoantigens usually display tissue-specific expression. In general, the work was found to be of excellent quality, and of broad interest.

Essential revisions:

Whereas an attempt to compare NLRP5 and KHDC3L autoantibodies was made, no such attempt to compare autoantibodies for TPH1 and RFX6 as markers for malabsorption/intestinal dysfunction. This would be a very valuable addition.

---

## [Author Response]

Essential revisions:Whereas an attempt to compare NLRP5 and KHDC3L autoantibodies was made, no such attempt to compare autoantibodies for TPH1 and RFX6 as markers for malabsorption/intestinal dysfunction. This would be a very valuable addition.

We agree that a comparison of novel anti-RFX6 antibodies to known anti-TPH1 antibodies would improve this manuscript, in particular by providing context for the frequencies of each of these antibodies as well as their relative associations with intestinal dysfunction (ID) subtypes – similar to what we provided for anti-NLRP5 versus anti-KHDC3L antibodies. To address this, we performed a whole protein radioligand binding assay for TPH analogously to the other RLBAs presented in this manuscript (Figure 6—figure supplement 3C). In terms of frequency, 55/67 APS1 sera were positive for anti-RFX6 antibodies, whereas only 45/67 had anti-TPH1 antibodies above the same cutoff, suggesting that anti-RFX6 antibodies may be more common than anti-TPH1 antibodies in APS1. Antibodies to both TPH1 and RFX6 were enriched in patients with intestinal dysfunction. However, in our cohort, anti-RFX6 antibodies tended to be enriched primarily in patients with diarrheal-subtype ID, while anti-TPH1 antibodies were distributed less specifically across both types (diarrheal- *and* constipation-type) of ID (Figure 6—figure supplement 3D). These observations are consistent with a previous report of anti-TPH1 antibodies showing an association with ID, but not specifically with the diarrheal subtype (Kluger, 2015). Overall, our anti-TPH1 data further support anti-RFX6 as the first reported APS1 autoantigen that correlates with diarrheal-type ID.

We have also added these comments to the Results section of the revised manuscript (subsection “Anti-enteroendocrine and anti-RFX6 response in APS1”, second paragraph).